# *Serenoa repens* and *Urtica dioica* Fixed Combination: In-Vitro Validation of a Therapy for Benign Prostatic Hyperplasia (BPH)

**DOI:** 10.3390/ijms21239178

**Published:** 2020-12-02

**Authors:** Miriam Saponaro, Isabella Giacomini, Giulia Morandin, Veronica Cocetta, Eugenio Ragazzi, Genny Orso, Ilaria Carnevali, Massimiliano Berretta, Mariangela Mancini, Francesco Pagano, Monica Montopoli

**Affiliations:** 1Department of Medicine, University of Padova, 35128 Padova, Italy; miriam.saponaro@vimm.it; 2Veneto Institute of Molecular Medicine, 35129 Padova, Italy; francesco.pagano1@gmail.com; 3Department of Pharmaceutical and Pharmacological Sciences, University of Padova, 35131 Padua, Italy; isabella.giacomini@studenti.unipd.it (I.G.); giulimorands@gmail.com (G.M.); veronica.cocetta@unipd.it (V.C.); eugenio.ragazzi@unipd.it (E.R.); genny.orso@unipd.it (G.O.); 4Clinical Research Department of Schwabe Pharma Italia, 39100 Bolzano, Italy; ilaria.carnevali@schwabe.it; 5Department of Medical Oncology-Centro di Riferimento Oncologico di Aviano (CRO), IRCCS, 33081 Aviano, Italy; berrettama@gmail.com; 6Urological Clinic, Department of Surgical, Oncological and Gastroenterological Sciences, School of Medicine and Surgery, University of Padova, 35124 Padova, Italy; mariangela.mancini@unipd.it

**Keywords:** BPH, inflammation, oxidative stress, *Serenoa repens*, *Urtica dioica*

## Abstract

Benign prostatic hyperplasia (BPH) is an age-related chronic disorder, characterized by the hyperproliferation of prostatic epithelial and stromal cells, which drives prostate enlargement. Since BPH aetiology and progression have been associated with the persistence of an inflammatory stimulus, induced both by Nuclear Factor-kappa B (NF-κB) activation and reactive oxygen species (ROS) production, the inhibition of these pathways could result in a good tool for its clinical treatment. This study aimed to evaluate the antioxidant and anti-inflammatory activity of a combined formulation of *Serenoa repens* and *Urtica dioica* (SR/UD) in an in vitro human model of BPH. The results confirmed both the antioxidant and the anti-inflammatory effects of SR/UD. In fact, SR/UD simultaneously reduced ROS production, NF-κB translocation inside the nucleus, and, consequently, interleukin 6 (IL-6) and interleukin 8 (IL-8) production. Furthermore, the effect of SR/UD was also tested in a human androgen-independent prostate cell model, PC3. SR/UD did not show any significant antioxidant and anti-inflammatory effect, but was able to reduce NF-κB translocation. Taken together, these results suggested a promising role of SR/UD in BPH and BPH-linked disorder prevention.

## 1. Introduction

Benign prostatic hyperplasia (BPH) is one of the most common phenomena related to the aging process, affecting 70–80% of men over the age of 80 [1]. Nevertheless, according to some epidemiological studies, BPH occurs also in half of the men between 50 and 60 years old [2,3]. BPH linkage to age is strictly due to the loss of prostatic function and secretory capacity, which are fundamental in young men for fertility and reproduction but spontaneously decrease during aging progression [1,4,5]. The hallmark of BPH is the hyperproliferation of prostatic epithelial and stromal cells, which drives prostate enlargement and, in most cases, the development of lower urinary tract symptoms (LUTS). Since LUTS often reduce patients’ life quality, it became of interest to find a treatment for BPH, which by itself is normally classified as a non-malignant disease [6,7].

One of the most important elements involved in BPH development is chronic inflammation. The critical role of pro-inflammatory cytokines and chemokines during BPH has been confirmed by several studies, suggesting also the use of these molecules as chronic prostate inflammatory biomarkers, potentially suitable for BPH diagnosis [8,9]. Particularly, the most relevant cytokines characterizing the prostate hyperplasic state are interleukin 6 (IL-6) and interleukin 8 (IL-8) [10,11].

Another important feature involved in BPH aetiology and pathology is oxidative stress. In fact, it has been reported that reactive oxygen species (ROS) play a dramatic role in sustaining the hyperproliferation of epithelial and stromal cells [12]. ROS production amplifies the recruitment of inflammatory cells, which in turn are responsible for producing a high amount of reactive species via the nicotinamide adenine dinucleotide phosphate (NADPH) pathway. Thus, oxidative stress is also responsible for the complication of the inflammatory state in the hyperplasic prostate [13].

Medicinal plants, in the form of plant parts or their extracts, are commonly used for the treatment of prostate diseases such as benign hypertrophy, prostatitis, and chronic pelvic pain syndrome. The pharmacological properties that are more interesting for the treatment of prostatic diseases are the anti-androgenic, anti-estrogenic, anti-proliferative, antioxidant, and anti-inflammatory ones [14].

In this work, two different drugs already approved for the treatment of BPH-associated inflammation, both containing the extract of *Serenoa repens*, have been compared with the aim to investigate the molecular mechanisms underlying their effects. The first one is a lipidosterolic extract of *Serenoa repens* (SR), while the second one consists of a combined formulation containing a fruit extract of *Serenoa repens* and a root extract of *Urtica dioica* (SR/UD). Particularly, the fruit extract of *Serenoa repens* has been indicated by several studies as a potential treatment for BPH [15,16,17,18]. The positive effect of this plant extract is due both to the direct inhibition of 5α-reductase and to the prevention of inflammation [19,20]. Recent meta-analyses showed that the effectiveness of *Serenoa repens* is similar or slightly inferior compared to finasteride and tamsulosin, but clearly higher than the placebo in the treatment of mild and moderate low urinary tract symptoms (LUTS), nocturia, and discomfort [21]. The combination with *Urtica dioica* has been proposed according to its anti-inflammatory and antioxidant activities [22,23]. In particular, we investigated whether the combination of *Urtica dioica* and *Serenoa repens* extracts permits the reduction of inflammation and oxidative stress in an in vitro cell model of prostatic hyperplasia (BPH-1 cell line).

Moreover, the same analyses conducted on BPH-1 cells were performed on the androgen-independent PC3 cell model in order to verify if the antioxidant and anti-inflammatory effects are similar in different prostate tissues or strictly linked to the BPH condition.

## 2. Results

### 2.1. Activity of SR and SR/UD in Cell Proliferation

Cell viability was determined to exclude the cytotoxicity of SR and SR/UD against BPH-1 and PC3 cells. Increasing concentrations (1, 10, 20 µg/mL) of both compounds were tested at different time points (24, 48, and 72 h). At each time point, concentration-response curves of cell viability were generated (Figure 1), but as expected, treatment with SR and SR/UD was not cytotoxic at the tested concentrations.

### 2.2. Antioxidant Effect of SR and SR/UD

It has been well established that increased ROS production leads to an imbalance of redox homeostasis. Moreover, oxidative stress is strictly linked to inflammation, as mentioned above [24,25]. Thus, in order to detect the potential antioxidant activity of SR and SR/UD against BPH-1 cells, ROS production assay was performed by using 2′,7′–dichlorofluorescein diacetate as previously described [26]. The antioxidant activity (Figure 2a–d) against BPH-1 cells treated with SR (1, 10, 20 µg/mL) or SR/UD (1, 10, 20 µg/mL) for 3 and 24 h was determined. SR/UD showed antioxidant activity, significantly decreasing ROS production, both after 3 h (1 and 10 µg/mL) (Figure 2a) and 24 h of treatment (Figure 2c). Particularly, SR/UD was able to counteract the increased ROS generation caused by the addition of the oxidant stimulus (H_2_O_2_) both after 3 h (1 µg/mL) (Figure 2b) and 24 h of treatment (Figure 2d). Treating cells with SR (1, 10 and 20 µg/mL), instead, increased ROS levels (Figure 2a–d).

The same analysis conducted on PC3 cells did not show antioxidant activity after both 3 and 24 h (Figure 2e–h).

As a positive control for antioxidant activity N-acetylcysteine (NAC), an established antioxidant [27], was used (Figure A1).

### 2.3. Anti-Inflammatory Effect of SR and SR/UD on BPH-1 Cells

The translocation of the transcriptional factor NF-kB in the nucleus was investigated in BPH-1 and PC3 cells treated with increasing concentrations of SR and SR/UD for 24 h. NF-κB translocation appeared to be decreased in BPH-1 cells after the treatment with SR and SR/UD at 10 µg/mL (Figure 3a,c).

In PC3 cells only, SR/UD (10 µg/mL) caused the reduction of NF-κB translocation in the cells’ nuclei (Figure 3b,d).

Since the regulation of inflammatory genes is strictly linked to the activation of NF-κB [28], we next tested the ability of SR (1, 10, 20 µg/mL) and SR/UD (1, 10, 20 µg/mL) to reduce the mRNA of some pro-inflammatory cytokines in lipopolysaccharides-stimulated BPH-1 and in PC3 cells. In fact, as already known, the overexpression of pro-inflammatory cytokines, such as IL-6 and IL-8, maintains chronic inflammation in the hyperplasic prostate [29,30,31].

In BPH-1 cells, IL-6 and IL-8 transcriptional levels were evaluated via quantitative polymerase chain reaction (qPCR) after treatment with SR and SR/UD as protocol. LPS treatment of BPH-1 cells significantly increased IL-6 and IL-8 transcription levels. On the contrary, pro-inflammatory cytokines appeared to be decreased by higher concentration treatments (Figure 4a,b). Particularly, IL-6 transcription decreased by 50% compared to LPS-treated cells after treatment with SR (1, 10, 20 µg/mL; *p* < 0.05) (Figure 4a). A slight down-regulation of IL-6 was shown by the treatment with SR/UD (10 and 20 µg/mL) (Figure 4a), which resulted in more efficiency in the modulation of IL-8 transcription. In fact, 10 and 20 µg/mL SR/UD treatments caused a decrease of 50 % of IL-8 levels (*p* < 0.05) (Figure 4b).

In PC3 cell analysis, the cytokine level was measured without LPS induction due to the high rate of inflammation-related genes under normal conditions [32]. In this case, treatments with SR and SR/UD were maintained for 72 h without adding the LPS after 48 h. Different from BPH-1 cells, PC3 treated with SR and SR/UD did not show alterations of IL-6 and IL-8 expression (Figure 4c,d).

The results obtained via qPCR for the expression of IL-6 and IL-8 were confirmed by the performance of enzyme-linked immunosorbent assays (ELISAs). In fact, Figure 5 shows that SR/UD has a strong effect in reducing the release of such cytokines by BPH-1 cells (Figure 5a,b). In PC3 cells, treatments with SR and SR/UD showed a lesser effect (Figure 5c,d).

## 3. Discussion

BPH is a common male disease strictly related to aging [4]. Several studies suggest the positive correlation between prostate aging and the development of a typical inflammatory microenvironment able to enhance cell proliferation and, in the long term, to contribute to the onset of BPH [9,33,34]. According to the critical role of chronic inflammation and ROS production in the pathogenesis and degeneration of BPH, targeting inflammatory mechanisms and oxidative stress could be a promising approach for the treatment of BPH and the consequent prevention of LUTS [35]. In fact, although the standard treatments for BPH consist of α-Adrenergic blockers and 5α-Reductase inhibitors [36,37,38,39], the role of plant extracts in BPH symptom counteraction has been recently demonstrated [37]. In our work, we investigated the antioxidant and anti-inflammatory activity of a drug approved for BPH treatment, SR/UD, in a human BPH in vitro model (BPH-1 cell line). SR/UD contains a patented formulation of *Serenoa repens* and *Urtica dioica*. Furthermore, we compared it with another approved drug for BPH treatment, consisting of an esanic fruit extract of *Serenoa repens* (SR). Previous studies on *Serenoa repens* proved its ability to ameliorate inflammation and to prevent ROS production in the BPH scenario [15,16,17,18]. *Urtica dioica* was indicated to possess anti-inflammatory activity as well [40]. Thus, our aim was to investigate for the first time the molecular mechanisms underlying the activity of the *Urtica dioica* and *Serenoa repens* combination, focusing on the antioxidant and anti-inflammatory properties.

The inflammatory status endurance, which is typical of BPH onset, is mostly triggered by the activation of NF-κB, one of the most important transcription factors involved in the inflammatory response [28]. The translocation of NF-κB inside the nucleus starts the transcription of different pro-inflammatory agents, such as IL-6 and IL-8 [8,41]. IL-6 is a proinflammatory cytokine that acts in the innate immune response [42], while IL-8 is a chemokine that promotes the migration of neutrophils, basophils, and T lymphocytes [43]. Thus, both of them are implicated in the recruitment of inflammatory cells and are consequently responsible for promoting BPH when overproduced over time [44,45]. Besides, the prostates of patients affected by BPH present higher IL-6 levels compared to the ones with a normal prostate [46]. The specific level of IL-6 is further recognized to be related to a poor outcome for these prostate diseases [47]. Several studies confirmed the direct correlation between IL-8 levels and BPH progression [48]. An important role in the persistence of the inflammatory microenvironment during BPH is also played by oxidative stress. ROS are continuously generated in the prostatic tissue as a result of hypoxia, which occurs after the development of abnormal blood flow patterns [49]. Our in vitro results show an antioxidant effect of SR/UD on BPH-1 cells. On the other hand, ROS are also responsible for the activation of NF-κB [50,51,52,53,54]. Our work highlights the significant downregulation of cytokines resulting from the reduced translocation of NF-κB inside the cells nuclei, suggesting the important anti-inflammatory activity of SR/UD. Thus, the patented combination of *Serenoa repens* and *Urtica dioica* plays an important role in the inhibition of the most important pro-inflammatory pathways involved in BPH development and progression directly through its antioxidant activity, which results in a decreased NF-κB-triggered inflammation. Accordingly, both the levels of IL-6 and IL-8 mRNA and the release of these cytokines appear to be diminished in the in-vitro model of BPH after the treatment with SR/UD. Particularly, IL-8 is the most affected by SR/UD.

The positive effect of SR/UD in the progression of prostatic diseases is also confirmed by previous clinical trials [55,56,57] and an in-vivo preclinical experiment [58]. Our in vitro results support the positive observed effect due to a deeper understanding of the molecular mechanism that underpins SR and SR/UD activity. Data show that treatment with SR/UD appears advantageous in the counteraction of oxidative stress and inflammation in BPH-1 cells. In particular, SR/UD has proven to be an effective antioxidant and this activity could be related to the prevention of the onset of inflammation, by reduction of NF-kB activation.

Considering that the aim of our work was to investigate the molecular mechanisms underlying the antioxidant and anti-inflammatory effects of SR/UD, both extracts were also tested in a human androgen-independent prostate cell model (PC3). SR/UD demonstrated the ability to reduce NF-κB translocation inside the nucleus and to slightly decrease the pro-inflammatory markers. Our results highlight the tissue-specificity of SR/UD, confirming its approved use in BPH treatment.

## 4. Conclusions

To conclude, the proven ability of SR/UD in diminishing the levels of ROS, pro-inflammatory interleukins, and translocated NF-κB confirms that its use is effective for the treatment of BPH. Finally, our results, taken together with the literature, suggest the hypothesis that ROS reduction is directly implicated in the decrease of the NF-κB inflammatory pathway and causes decreased secretion of inflammatory cytokines. Consequently, the reduction of inflammation contributes to mitigating prostate gland enlargement.

## 5. Materials and Methods

### 5.1. Cell Cultures

The benign prostatic hyperplasia cell line (BPH-1) was purchased from Leibniz Institute DSMZ-German Collection of Microorganisms and Cell Cultures GmbH (Braunschweig, Germany) and cultured in Roswell Park Memorial Institute medium (RPMI 1640) (Gibco, Thermo Fisher Scientific, Waltham, MA, USA) supplemented with 20% h.i. FBS (Gibco, Thermo Fisher Scientific, Waltham, MA, USA), 20 ng/mL DHT (Supelco, Merck, Darmstadt, Germany), 5 µg/mL transferrin (Sigma–Aldrich, Merck, Darmstadt, Germany), 5 ng/mL sodium selenite (Sigma–Aldrich, Merck, Darmstadt, Germany) and 5 µg/mL insulin (Sigma–Aldrich, Merck, Darmstadt, Germany). The prostate cancer 3 cell line (PC3) was obtained by ATCC (Manassas, VA, USA) and grown with RPMI 1640, 10% FBS, 2 mM glutamine, 100 U/mL penicillin, and 100 µg/mL streptomycin. Both cell lines were maintained under standard culture conditions at 37 °C and 5% CO_2_. Cells were collected every 2 days with a minimum amount of 0.05% trypsin–0.02% EDTA.

### 5.2. SR Composition

Lipido sterolic extract from fruits of *S. repens* [320 mg *S. repens*; drug–extract ratio (7–11:1); extraction solvent: hexane].

### 5.3. SR/UD Composition

WS^®^ 1541, the active portion of SR/UD, is a fixed combination of:lipophilic extract from fruits of *S. repens* [160 mg WS^®^ 1473; drug–extract ratio 10.0–14.3:1; extraction solvent: 90% ethanol (m/m)]aqueous ethanolic extract from roots of *U. dioica* [120mg WS^®^ 1031, drug–extract ratio 7.6–12.5:1; extraction solvent: 60% ethanol (m/m)].

### 5.4. Cells Viability Assay

Crystal violet assay was used to assess cell viability after treatment with SR/UD or SR. BPH-1 and PC3 cells were seeded at respective densities of 15,625 × 10^3^ cells/cm^2^ and 31,125 × 10^3^ cells/cm^2^ in 96-well plates and treated after 24 h with different concentrations (1, 10, 20 µg/mL) of both SR and SR/UD. The crystal violet assay was performed after 1, 2, and 3 days of treatment. Cells were firstly fixed with 2% formaldehyde (Sigma–Aldrich, Merck, Darmstadt, Germany) for 15 min, washed twice with PBS without bivalent cations (Euroclone, Milan, Italy), and stained with 0.1% Crystal violet solution (Sigma–Aldrich, Merck, Darmstadt, Germany) for 20 min. They were washed three times with PBS and dried overnight. Then, 10% acetic acid (Sigma–Aldrich, Merck, Darmstadt, Germany) was used to lyse the stained cells, and the absorbance was measured at 570 nm using a Victor3X multilabel plate counter (Wallac Instruments, Turku, Finland). Growth curve analysis was carried out according to Ishiyama [59].

### 5.5. ROS Assay

BPH-1 and PC-3 cells (5 × 10^3^ cells/well) were seeded in 96-well plates. After 24 h, cells were treated with SR and SR/UD (1, 10, 20 µg/mL) for 3 or 24 h and incubated at 37 °C. Then, cells were incubated with a 100 µM diacetylated 2′,7′-dichlorofluorescein (DCF-DA) probe (Sigma–Aldrich, Merck, Darmstadt, Germany) for 30 min at 37 °C, in the presence or absence of H_2_O_2_ 0.9 µM (Sigma–Aldrich, Merck, Darmstadt, Germany), and the fluorescence was measured by using a Victor3X multilabel plate counter (Ex 485 nm and Em 535 nm) (Wallac Instruments, Turku, Finland). This assay takes advantage of the fluorescence emitted by the oxidation of the non-fluorescent DCF-DA and evaluates intracellular ROS production.

### 5.6. Nuclear Factor-kappa B Translocation Assay

To perform the NF-κB translocation assay, BPH-1 and PC3 cells were firstly seeded on glass coverslips in 24-well plates and cultured as previously indicated. When cells reached approximately 30% confluence, they were fixed with 4% formaldehyde (Sigma–Aldrich, Merck, Darmstadt, Germany), permeabilized with 0.1% Triton X-100 (Sigma–Aldrich, Merck, Darmstadt, Germany) in PBS, and stained 1 h with rabbit monoclonal anti-NF-κB p65 (Invitrogen, Life Technologies, Milan, Italy). After PBS washes, they were incubated with anti-rabbit secondary antibody Alexa Fluor 488 (Molecular Probes, Invitrogen, Milan, Italy) for 1 h at room temperature. Cells were then washed with PBS and stained with Hoechst (1:10,000) (Invitrogen, Life Technologies, Milan, Italy). The coverslips were finally mounted on glass slides by using Mowiol 40–88 (Sigma–Aldrich, Merck, Darmstadt, Germany). Images were acquired through a 60x CFI Plan Apochromat Nikon objective with a Nikon C1 confocal microscope and finally analysed using NIS Elements software (Nikon Instruments, Florence, Italy), NIH Image J version 1.52t, and Adobe Photoshop CS4 version 11.0.2 (Adobe, San Jose, CA, USA).

### 5.7. RNA Expression/Quantitative Real-Time PCR

BPH-1 and PC-3 cells were seeded at respective densities of 15,625 × 10^3^ cells/cm^2^ and 31,125 × 10^3^ cells/cm^2^ in 6-well plates. BPH-1 cells were pre-treated with SR or SR/UD (1, 10, 20 µg/mL) and then stimulated with LPS (10 µg/mL) (Sigma–Aldrich, Merck, Darmstadt, Germany) for 24 h. The treatments with SR and SR/UD were maintained during the LPS stimulation. The duration of the SR and SR/UD treatments was a total of 72 h. PC3 cells were treated for 72 h with SR and SR/UD, avoiding the stimulation with LPS. RNA was isolated with the TRIzol method (Invitrogen, Thermo Fisher Scientific, Waltham, MA, USA), and retro-transcription was performed with a High-Capacity cDNA Reverse Transcription Kit (Applied Biosystems, Thermo Fisher Scientific, Waltham, MA, USA) according to the manufacturer’s instructions. Quantitative PCR (qPCR) reactions were performed using QuantStudio™ 5 (Invitrogen, Thermo Fisher Scientific, Waltham, MA, USA) with Power SYBR™ Green PCR Master Mix (Applied Biosystems, Thermo Fisher Scientific, Waltham, MA, USA) and the specific primers reported below. Primer sequences were obtained from PrimerBank (http://pga.mgh.harvard.edu/primerbank/index.html). The GAPDH expression level was used as a reference for the normalization of each value level. The primer sequences used were as follows: IL-6 forward, 5′-TACATCCTCGACGGCATCTC-3′; reverse, 5′-TGCCTCTTTGCTGCTTTCAC-3′. IL-8 forward, 5′-TTGGCAGCCTTCCTGATTTC-3′; reverse, 5′-TTGGGGTGGAAAGGTTTGGAG-3′. GAPDH forward 5′-AATCCCATCACCATCTTCCA-3′; reverse, 5′-TGGACTCCACGACGTACTCA-3′.

### 5.8. IL-6 and IL-8 ELISA

BPH-1 and PC-3 cells (50 × 10^3^ cells/well) were seeded in 6-well plates. Cells were pre-treated with SR or SR/UD (10, 20 µg/mL) for 24 h and stimulated with LPS (10 µg/mL) for 6 h. The media were then replaced with fresh media without FBS, which were collected after 12 h for the detection of IL-6 and IL-8 levels. Human IL-6 (limit of sensitivity is 3 pg/mL) and IL-8 (limit of sensitivity is 4 pg/mL) levels were determined by an ELISA kit (RayBio, Peachtree Corners, GA, USA and BioLegend, San Diego, CA, USA respectively) according to the protocols. Results are expressed as pg/mL and reported as the means of three independent experiments.

### 5.9. Statistical Analysis

GraphPad Prism version 3.03 (GraphPad, San Diego, CA, USA) software was used for statistical evaluation of the experimental results. Data were analyzed using one-way analysis of variance (ANOVA), followed by an appropriate *post-hoc* test (Bonferroni or Tukey-Kramer). A *p* value of less than 0.05 was considered for determining statistical significance.

## Figures and Tables

**Figure 1 ijms-21-09178-f001:**
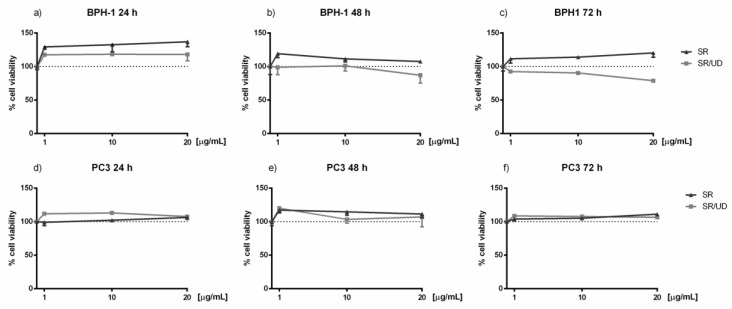
Effect of SR and SR/UD on BPH-1 and PC3 cell viability. BPH-1 cells treated with increasing concentrations of SR and SR/UD (1, 10, 20 µg/mL) for (**a**) 24 h, (**b**) 48 h, (**c**) 72 h. PC-3 cells treated with increasing concentrations of SR and SR/UD (1, 10, 20 µg/mL) for (**d**) 24 h, (**e**) 48 h, (**f**) 72 h. BPH-1 and PC3 cell viability was evaluated with the Crystal Violet Assay. Data represent the mean ± SEM (*n* = 3). Differences between various drug concentrations and control were analysed for each time point (24, 48, 72 h). The dashed line is the control.

**Figure 2 ijms-21-09178-f002:**
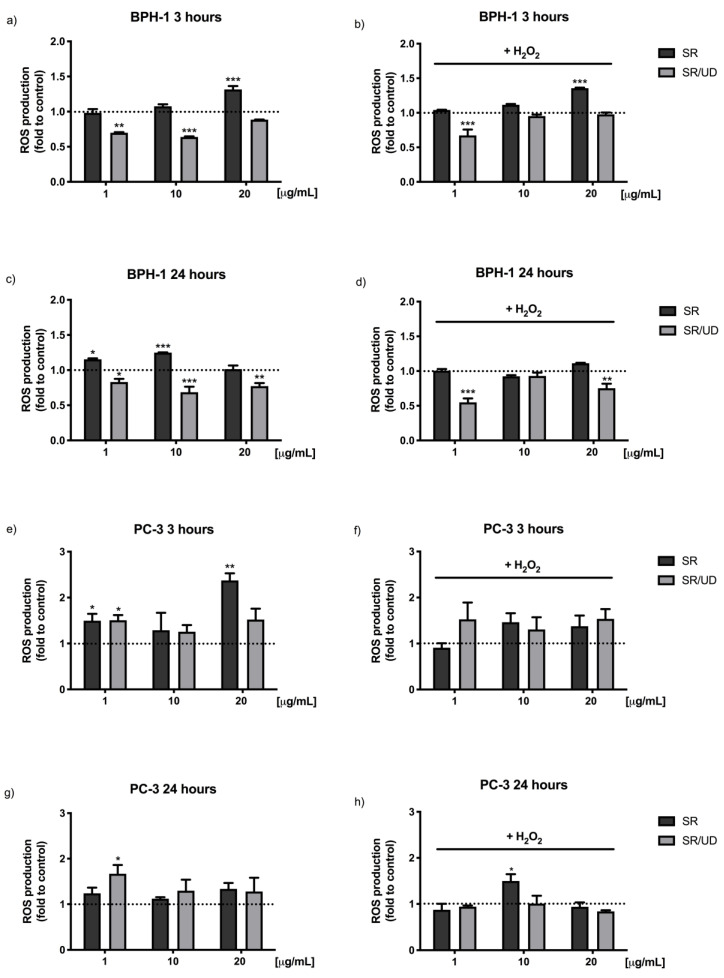
Effect of SR and SR/UD (1, 10, 20 µg/mL) on ROS generation in BPH-1 and PC3 cells. The cells were treated for 3 h (**a**,**b**,**e**,**f**) or for 24 h (**c**,**d**,**g**,**h**) with the two compounds. ROS were detected by DCF-DA staining at the basal condition (**a**,**c**,**e**,**g**) and following the exposure to oxidative stimulus H_2_O_2_ (**b**,**d**,**f**,**h**). The fluorescence was measured by using a Victor3X multilabel plate counter (Ex 485 nm and Em 535 nm). Results are expressed as the fold change of ROS production. Each bar represents the mean ± SEM (*n* = 3). * *p* < 0.05, ** *p* < 0.01, *** *p* < 0.001, treatment compared to control. The dashed line is the control.

**Figure 3 ijms-21-09178-f003:**
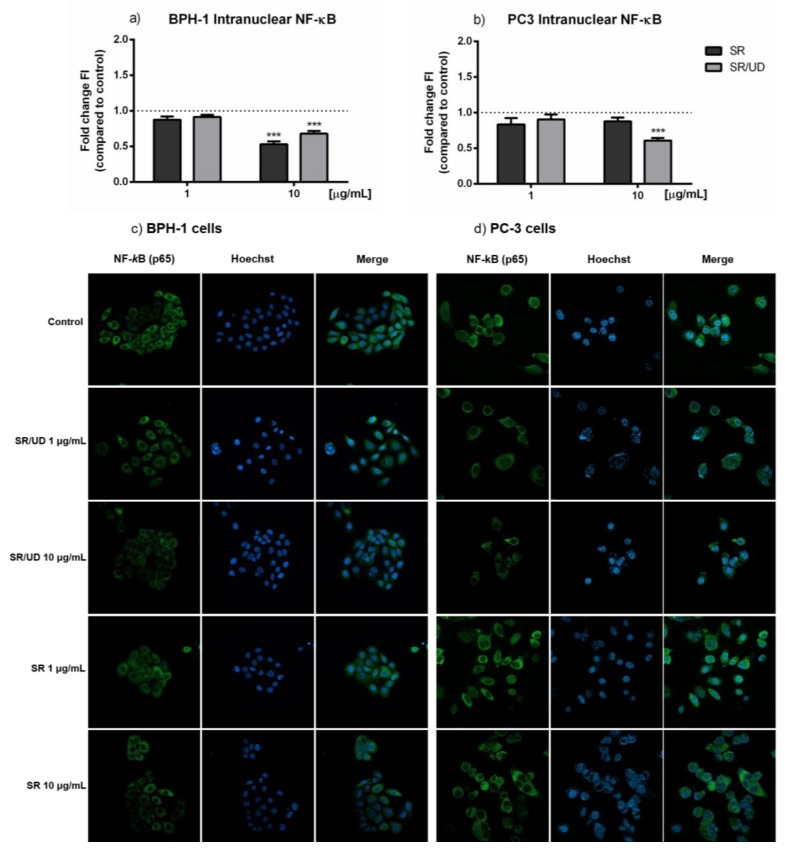
Effect of 24 h treatments with SR and SR/UD (1, 10 µg/mL) on nuclear translocation of NF-κB in BPH-1 and PC3 cells. (**a**,**b**) Quantification of intranuclear NF-KB in BPH-1 cells (**a**) and PC3 cells (**b**). Values are expressed as the ratio between treatments and control. Data represent the mean ± SEM (*n* = 3). *** *p* < 0.001, treatment compared to the control (indicated as the dashed line). (**c**,**d**) Representative images of NF-κB immunofluorescence staining in BPH-1 cells (**c**) and PC3 cells (**d**) treated with SR and SR/UD. Images were captured at 60X magnification.

**Figure 4 ijms-21-09178-f004:**
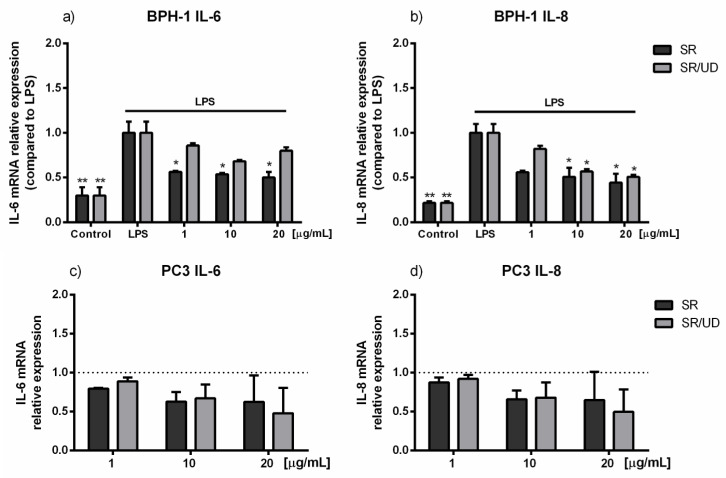
Effect of SR and SR/UD (1, 10, 20 µg/mL) on IL-6 and IL-8 mRNA expression in BPH-1 and PC3 cells. (**a**) IL-6 expression in BPH-1 cells pre-treated with SR and SR/UD for 48 h and stimulated with LPS (10 µg/mL) for 24 h. (**b**) IL-8 expression in BPH-1 cells pre-treated with SR and SR/UD for 48 h and stimulated with LPS (10 µg/mL) for 24 h (**c**) IL-6 expression in PC3 cells treated with increasing concentrations of SR and SR/UD for 72 h. (**d**) IL-8 expression in PC3 cells treated with increasing concentrations of SR and SR/UD for 72 h. IL-6 and IL-8 were evaluated by qPCR. IL-6 and IL-8 are expressed as the ratio between treatments and LPS stimulus * *p* < 0.05, treatment compared to LPS, ** *p* < 0.01, Control compared to LPS. Data represent the mean ± SEM (*n* = 3).

**Figure 5 ijms-21-09178-f005:**
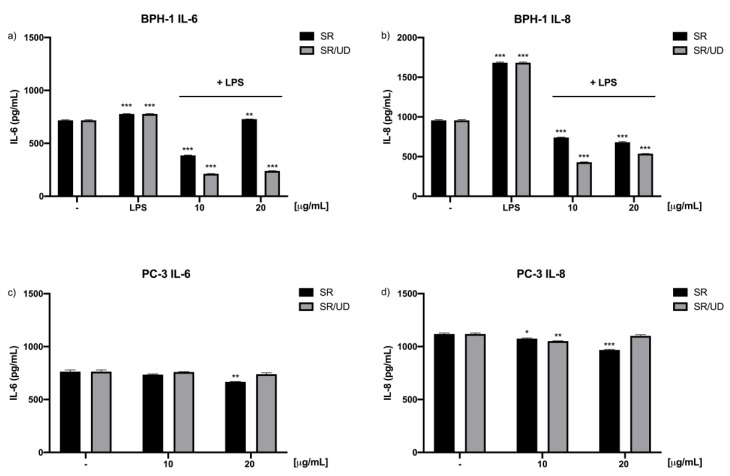
Human IL-6 and IL-8 production (pg/mL) in BPH-1 and PC3 cells. (**a**) IL-6 production by BPH-1 cells pre-treated with SR and SR/UD (10, 20 µg/mL) for 24 h and stimulated with LPS (10 µg/mL) for 6 h. (**b**) IL-8 production by BPH-1 cells pre-treated with SR and SR/UD (10, 20 µg/mL) for 24 h and stimulated with LPS (10 µg/mL) for 6 h. (**c**) IL-6 expression by PC3 cells treated with SR and SR/UD (10, 20 µg/mL) for 24 h. (**d**) IL-8 expression by PC3 cells treated with SR and SR/UD (10, 20 µg/mL) for 24 h. IL-6 and IL-8 are expressed as the ratio between treatments and LPS stimulus. * *p* < 0.05, ** *p* < 0.01, *** *p* < 0.001, LPS stimulus compared to the control or treatment compared to the control. Data represent the mean ± SEM (*n* = 3).

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
