# Peer review of "Serenoa repens and Urtica dioica Fixed Combination: In-Vitro Validation of a Therapy for Benign Prostatic Hyperplasia (BPH)"

_ijms, 2020, doi:10.3390/ijms21239178_

Round 1

Reviewer 1 Report

The authors aimed to perform an in-vitro validation of  serenoa repens and urtica dioica for the treatment of benign prostatic  hyperplasia.

However, the aims  and the findings are not so much in line with the results and conclusions. Moreover, some sentences must be better exposed since they could be considered unclear.

E.g.:

Line 44: the sentence is misleading, the conclusions of the study (ref. 8) are different from this. Please be more careful and change the sentence.

Also the sentence “Both benign and malignant prostate cancers are characterized by chronic inflammation, which is commonly reported also in BPH” has to be change since the existence benign prostate cancers is at least “of debate”

Line 77: … permits the reduction of inflammation and oxidative stress caused by BPH… It is not correct: inflammation and oxidative stress drive BPH not the contrary

The introduction is based on speculations about other papers results more than the exact report of solid finding. The authors are presenting preliminary results or little findings that still need  confirmation as a fact, perplexing the reader.

Moreover I can’t understand why a lipidic extract is compared to fruit extract. Indeed, liposolubility is crucial for prostate drugd penetration.

Please specify LPS.

A more clear discussion, with less speculations is needed. Moreover, I suggest to refer to consistent findings, above all about the links between inflammation, BPH and PCa.

Author Response

The authors aimed to perform an in-vitro validation of  serenoa repens and urtica dioica for the treatment of benign prostatic  hyperplasia.

However, the aims  and the findings are not so much in line with the results and conclusions. Moreover, some sentences must be better exposed since they could be considered unclear.

The authors are grateful to Reviewer 1 for her/his valuable comments. The manuscript has been revised following all the major points suggested.

Here below in red are our answers to each raised point.

E.g.:

Line 44: the sentence is misleading, the conclusions of the study (ref. 8) are different from this. Please be more careful and change the sentence.

Also the sentence “Both benign and malignant prostate cancers are characterized by chronic inflammation, which is commonly reported also in BPH” has to be change since the existence benign prostate cancers is at least “of debate”

Lines 44-51: the sentences have been changed with “Moreover, it is well known that BPH shares multiple predisposing factors with prostate cancer (PC), such as inflammation, oxidative stress, hormone imbalances, metabolic and genetic factors [8–14]. Particularly, several studies confirm the role of inflammation in high-grade and low-grade PC development [15–17].

Line 77: … permits the reduction of inflammation and oxidative stress caused by BPH… It is not correct: inflammation and oxidative stress drive BPH not the contrary

Lines 81-85: the sentence has been changed with “In particular, we investigated whether the addition of the root extract of Urtica dioica to Serenoa repens extract in an in vitro cell model of prostatic hyperplasia (BPH-1 cell lines) permits the reduction of inflammation and oxidative stress, exploring how this natural mixture is able to arrest in vivo the progression of BPH and to avoid the onset of its symptoms”.

The introduction is based on speculations about other papers results more than the exact report of solid finding. The authors are presenting preliminary results or little findings that still need  confirmation as a fact, perplexing the reader.

The authors thank the reviewer for this comment. They modified the introduction in order to avoid the speculations’ excess. The literature on this field is still poor, but the aim is to better understand and improve the quality of the actual knowledge.

Moreover I can’t understand why a lipidic extract is compared to fruit extract. Indeed, liposolubility is crucial for prostate drugd penetration.

The lipidic esanic extract and the ethanolic extract from Serenoa’s fruits are the only two extract mentioned in the EMA monograph for Well Estabilished Use drugs and for Traditional Use drugs.

Thus, the authors considered interesting comparing only these two types of extract.

In addition, the extraction with ethanol is suitable to obtain an extract rich in lipofilic compounds, in order to guarantee prostate penetration.

EMA itself evaluated the efficacy of ethanolic extract in the specific Assesment, reporting many clinical data.

https://www.ema.europa.eu/en/documents/herbal-report/final-assessment-report-serenoa-repens-w-bartram-small-fructus_en.pdf

Please specify LPS.

In the main text, LPS has been specified as suggested.

A more clear discussion, with less speculations is needed. Moreover, I suggest to refer to consistent findings, above all about the links between inflammation, BPH and PCa.

Lines 276-286: “To conclude, the proven ability of SR/UD in diminishing the levels of ROS, pro-inflammatory interleukins, NALP3, and translocated NF-κB confirms it as a natural compound that can be used effectively for the treatment of BPH. Moreover, this study demonstrates that the addition of Urtica Dioica in the extract formulation enhances the antioxidant and anti-inflammatory activity of Serenoa Repens. Finally, our results, taken together with the literature, suggest the hypothesis that ROS reduction is directly implicated in the decrease of NF-κB translocation, which in turn allows a lower production of pro-inflammatory cytokines and inflammasome components. Consequently, the reduction of inflammation contributes to mitigating prostate gland enlargement. Moreover, the slighter deregulation of the pro-inflammatory pathways observed in the in-vitro PC model suggests the possibility to undertake a deeper investigation of the effects of Serenoa Repens and Urtica Dioica over PC prevention and treatment. ”

Reviewer 2 Report

The authors have determined the combined effect of SR/UD as anti-inflammatory in BPH-1 and PC-3 prostate cells. The results are interesting but more experiments are required before acceptance for publication.

  1. The experimental design does not allow to evaluate if there are synergistic effects between SR and UD. For example, in fig.1, no effects are observed with SR alone and the combination SR/UD decreases the ROS production, and thus, this effect can only be due to the effect of UD. Experiments using the effect of UD alone are needed.
  2. The levels of IL-6 and IL-8 production should be determined by ELISA and using earlier time points. The result of qPCR at 72h can be due to a negative feedback and does not represent the real production and secretion of ILs.
  3. The NLRP3 protein should be determined by western blot.
  4. The immunoflurescence image should be added to figure 5.
  5. What is present on extracts that can exlplain the results? The chemical characterization should be carried out using, for example, LC-MS. These results should be dicussed.
  6. The number of figures is too long. You can join in the same figure the BPH-1 and PC-3 cells.
  7. The discussion should be updated in according with new results.

Author Response

The authors have determined the combined effect of SR/UD as anti-inflammatory in BPH-1 and PC-3 prostate cells. The results are interesting but more experiments are required before acceptance for publication.

The authors are grateful to Reviewer 2 for her/his valuable comments. The manuscript has been revised following all the major points suggested.

Here below in red are our answers to each raised point.

  1. The experimental design does not allow to evaluate if there are synergistic effects between SR and UD. For example, in fig.1, no effects are observed with SR alone and the combination SR/UD decreases the ROS production, and thus, this effect can only be due to the effect of UD. Experiments using the effect of UD alone are needed.

The evaluation among UD extract alone, SR extract alone and the combination UD and SR was not the aim of this work, because in the italian market it doesn’t exist a product containing only SR extract or UD extract alone.

We would like to compare the effects of two extracts, really present in different products in the italian market.

  1. The levels of IL-6 and IL-8 production should be determined by ELISA and using earlier time points. The result of qPCR at 72h can be due to a negative feedback and does not represent the real production and secretion of ILs.

The levels of IL-6 and IL-8 have been determined by ELISA. Cells were pre-treated for 24 hours and stimulated for 6 hours with LPS.

3. The NLRP3 protein should be determined by western blot.

The authors did not have enough time to perform the western blot because the antibody that they ordered did not arrive, considering also the Covid-19 delivery’s restrictions.

4. The immunoflurescence image should be added to figure 5.

The immunofluorescence images have been added. Now they are in the text as Figure 6.

5. What is present on extracts that can exlplain the results? The chemical characterization should be carried out using, for example, LC-MS. These results should be dicussed.

WSR 1473 saw palmetto extract

Extracts of saw palmetto fruits contain saturated and unsaturated fatty acids which are present in their free form, but also bound as ethyl ester and triglycerides.

Other constituents of the extracts are free and conjugated phytosterols, various other lipids, fatty oils, and resins. [Koch 1995]

 WSR 1031 dry stinging nettle root extract

Extracts of stinging nettle roots contain phytosterols, triterpene derivatives, phenylpropane derivatives (e. g. lignanes), ceramides, hydroxy fatty acids,

polysaccharides, simple phenolic compounds, and lectins. [Koch 1995]

 WS 1473 and WS 1031 are both patented extracts, and their detailed composition has not been disclosed by the company that owns the patent

6. The number of figures is too long. You can join in the same figure the BPH-1 and PC-3 cells.

The authors thank the reviewer for this suggestion and they joined in the same figure BPH-1 and PC-3.

7.The discussion should be updated in according with new results.

The discussion has been updated with the new results.

Round 2

Reviewer 1 Report

Some concerns are still present after revision:

Reference 15-17 don't refer to prostate cancer but PIN or PIA that are not cancer. So the statement is incorrect. Moreover, there is no evidence that prostae cancer is linked with the progression of BPH. I ask to better check the consistency of literature to make these affirmations. Please provie a simple description and discussion of your in vitro results not too much speculating on what will happen in vivo in BPH or PCa patients, since it is not evaluated in your paper. Please check your references since some of the papers you referred don't precisely say what you wrote in the text.  

Author Response

Responses to Reviewer 1

Some concerns are still present after revision: Reference 15-17 don't refer to prostate cancer but PIN or PIA that are not cancer. So the statement is incorrect. Moreover, there is no evidence that prostae cancer is linked with the progression of BPH. I ask to better check the consistency of literature to make these affirmations. Please provie a simple description and discussion of your in vitro results not too much speculating on what will happen in vivo in BPH or PCa patients, since it is not evaluated in your paper. Please check your references since some of the papers you referred don't precisely say what you wrote in the text.  

The authors are grateful to Reviewer 1 for her/his valuable comments. The manuscript has been revised following all the major points suggested.

The authors checked carefully all the references, adjusting the mistakes and misleading sentences. Moreover, we rearrange the main body and the discussion of the manuscript focusing on the literature and on the actual state of art of the topic to support our data, without inferencing or speculating on in vivo results.

Reviewer 2 Report

I understand the asnswers of authors, but technically, if you want to compare the effects triggered by SR alone and combined effect of SR/UD you have to carry out experiments using the extracts alone as control. You can easly find the extracts of UD in international market. You can ask to editor additional time to carry out the experiments.

If the goal is only to evaluate the combined effect in oxidative stress and inflammation you should reorganize the article with only those results. 

Author Response

Responses to Reviewer 2

I understand the asnswers of authors, but technically, if you want to compare the effects triggered by SR alone and combined effect of SR/UD you have to carry out experiments using the extracts alone as control. You can easly find the extracts of UD in international market. You can ask to editor additional time to carry out the experiments.

If the goal is only to evaluate the combined effect in oxidative stress and inflammation you should reorganize the article with only those results. 

The authors thank the reviewer 2 for the comments. We worked to address the major points suggested.

The main goal of our study is to demonstrate the antioxidant and anti-inflammatory effect of the combined formulation. Thus, we reorganize the results section of our article, pointing the attention on the efficacy of the combination and not on the comparison with the other drug. We maintained the data about SR alone since it is of interest to have data about the activity of another approved drug for treatment of the same BPH pathology.

Round 3

Reviewer 1 Report

The paper is ready for publication

Reviewer 2 Report

The article can be accepted for publication.